# Temperature Measurements by Wavelength Modulation Diode Laser Absorption Spectroscopy with Logarithmic Conversion and 1*f* Signal Detection

**DOI:** 10.3390/s23020622

**Published:** 2023-01-05

**Authors:** Vladimir Liger, Vladimir Mironenko, Yury Kuritsyn, Mikhail Bolshov

**Affiliations:** Institute of Spectroscopy, Russian Academy of Sciences, 5 Fizicheskaya Str., Moscow 108840, Russia

**Keywords:** tunable diode laser absorption spectroscopy, TDLAS, temperature measurements, multiplicative noise, wavelength modulation, first harmonic, logarithmic processing

## Abstract

A new version of a sensor for temperature measurements in the case of strong laser intensity fluctuation was developed. It was based on tunable diode laser absorption spectroscopy (TDLAS) with wavelength modulation, logarithmic conversion of the absorption signal, and detection of the first harmonic of the modulation frequency. The efficiency of the technique was demonstrated under experimental conditions with excess multiplicative noise. Temperature was evaluated from the ratio of integrated absorbance of two lines of the water molecule with different lower energy levels. Two algorithms of data processing were tested, simultaneous fitting of two spectral ranges with selected absorption lines and independent fitting of two absorption lines profiles. The correctness of the gas temperature evaluation was verified by simultaneous measurements with a commercial thermocouple. An error in temperature evaluation of less than 40 at 1000 K was achieved even when processing a single scan of the diode lasers.

## 1. Introduction

A variety of techniques have been developed and used for temperature measurements in combustion systems [1,2,3,4]. There are non-optical (thermocouples) and optical diagnostics (absorption spectroscopy, imaging, laser-induced fluorescence, and coherent anti-Stokes–Raman scattering CARS). The results of such diagnostics of a hot zone are used both in fundamental investigations and technological applications. 

The decisive advantage of optical methods is their “non-invasiveness”. The intensity of the probing radiation is so low (on the order of a few mW) that it does not affect the processes in the probed zone. In contrast, the use of thermocouples to determine *T* is fundamentally impossible, since their insertion into the test zone significantly affects the values and spatial distribution of the parameters of the hot jets or internal combustion chambers with temperatures from some hundreds to some thousands K. In many cases the diagnostic of such types of media needs temporal resolution in the ms-µs range, which is impossible for thermocouples but can be realized by laser-based spectroscopic techniques.

Due to the relative simplicity and high sensitivity in detecting atoms and molecules, tunable diode laser absorption spectroscopy (TDLAS) has received special attention. In hot zones, the TDLAS technique allows for the measurement of temperature, total pressure of the gas mixture, and partial pressures of the main molecular components with a time resolution in the micro–millisecond range [5,6,7,8]. The main advantages of the TDLAS technique include non-intrusive, non-perturbing measurements, the relative simplicity of design, the relatively low cost of the main components, and the possibility of delivering the probing radiation of a diode laser (DL) to the hot zone via optical fiber. This last advantage makes it possible to locate the sensitive recording part of the TDLAS sensor away from the testing hot zone with high acoustic and electrical noise [9]. 

The method of determining the temperature of a gas medium using TDLAS is based on the measurement of the integrated absorbance on several lines of a test molecule having different lower energy levels [10]. The lines’ integrated absorbances, in turn, are determined through a process of fitting a simulated spectrum, constructed using spectroscopic databases, to the measured one. Having determined the temperature of the medium in this way, it is also possible to find the concentration of the molecular components of a mixture by measuring the absorption of the probing DL radiation in the medium.

To date, various versions of TDLAS have been developed for determining the temperature: direct absorption spectroscopy (DAS) and several variants of wavelength-modulation spectroscopy (WMS). It should be noted that in these methods, the absorption of a test molecule is measured against a baseline, determined by different sources. The recorded signal is proportional to DL intensity, and, thus, depends on the time-varying DL intensity and its fluctuations.

DAS is the simplest method for determining the parameters of a medium [11,12,13]. It is easily interpreted, and works well in cases when the noise is small compared to the magnitude of the absorption signal. In industrial applications, especially in combustion installations with supersonic jets, DAS has limited use due to strong noise distorting the absorption signal. Such noises are both additive in nature (broadband radiation of a hot zone, electromagnetic pick-ups, etc.), and multiplicative, caused by fluctuations in the intensity of laser radiation transmitted through the measuring region. This situation is typical for strong turbulence in the gas zone, in the presence of scattering particles in the laser beam path, and high levels of acoustic and electromagnetic noise.

To significantly reduce the effect of additive noise on the accuracy of determining the intensity of absorption lines, WMS (wavelength modulation spectroscopy) is used [14,15,16,17,18,19]. In this technique, in addition to slow scanning of the DL wavelength across the absorption line (with frequencies of the order of 1 kHz), a fast modulation of the wavelength is applied with a modulation amplitude comparable to the width of the absorption line and with frequencies *f* of the order of 10–100 kHz. The absorption signal is detected at harmonics of *kf* (usually 2*f*) using a lock-in-amplifier (LIA). If the frequency of *kf* is outside the spectrum of additive noise, then the latter can be greatly attenuated, which leads to an increase in the signal-to-noise ratio compared to DAS. Measurements at higher (*k* > 1) harmonics also make it possible to reduce the influence of the laser radiation residual amplitude modulation (RAM) accompanying the wavelength modulation. However, the WMS signal when registered at one harmonic (as well as the DAS signal), due to its multiplicative nature, depends on the intensity of the laser radiation being recorded.

This difficulty was overcome by normalizing the *kf* harmonic of the signal to the first harmonic [20,21,22,23]. The signal received as a result of normalization does not depend on the intensity of laser radiation. The signal processing, taking into account the tuning characteristics of the DL, improves the accuracy of a temperature determination in a hot zone. It should be noted that in any version of the WMS method, data on the parameters of the gas under test can be obtained only after determination of the tuning and modulation characteristics of the DLs. The algorithms for processing raw data in the WMS method are much more complex than in the DAS method.

One of the radical ways to eliminate the dependence of the sensor output signal on non-selective variations in the intensity of the detected laser radiation is the use of a logarithmic conversion. Several applications of logarithmic conversion in conjunction with WMS log-WMS have been described. In [24,25], when detecting Cl atoms in a glow discharge, the log-WMS technique provided suppression of excessive laser noise. The increase in linear dynamic range of the output signal using log-WMS was demonstrated in [26]. The log-WMS technique was used to determine the concentration of absorbing molecules [27,28,29,30]. However, all studies used the registration of the second harmonic at a relatively low frequency of laser modulation, and measurements were carried out on separate isolated absorption lines to determine the concentration of the tested atoms or molecules.

In our work [31], a version of the log-WMS technique was proposed using a combination of logarithmic conversion of the experimental data and modulation spectroscopy with the registration of the first harmonic of the absorption signal of the test molecule (log-WMS-1*f*). The proposed version of the TDLAS technique and the processing algorithm significantly simplify the procedure for the evaluation of the experimental data in comparison with the published versions of the WMS. In [31], the proposed technique was demonstrated by determining the integrated absorbance of a single absorption line of an H_2_O molecule at atmospheric pressure and room temperature. 

In this paper, the possibility of using the technique developed in [31] to determine the temperature of a hot zone by measuring the first harmonic of the signal of two absorption lines of a test molecule was investigated. Two DLs were used in the range of 1.3–1.4 µm. Scanning of the laser wavelengths in the vicinity of the selected absorption lines of H_2_O was carried out alternately (time multiplexing) with a frequency of 122 Hz. At the same time, the wavelengths of the DLs were modulated with a frequency of ~50 kHz. The operation of the proposed technique under conditions of strong multiplicative noise and broadband thermal radiation was investigated in detail. Data collection and signal processing algorithms were based on the logarithmic conversion of DL signals, a differential registration scheme, and LIA at the first harmonic of the modulation frequency. The temperatures inferred from absorption measurements were compared with thermocouple data in the presence of multiplicative noises and without them. The influence of background radiation, which after logarithmic conversion causes an additive signal, was analyzed in detail. The advantages and limitations of the proposed technique are discussed.

## 2. Theoretical Background

The basic equation of absorption spectroscopy is the Beer-Lambert law. For an isolated line, it can be expressed in the form: (1)$$\frac{I_{t}}{I_{0}}=\exp\left[-\alpha \left(\nu \right)\right]=\exp\left[-\int_{0}^{L}S\left(T\right)Ng\left(\nu \right)dl\right]$$
where *I_t_* is the intensity of the monochromatic radiation with a frequency *ν* (cm^−1^) transmitted through an absorbing medium of length *L* (cm), *I*_0_ is the intensity of the incident radiation, *α*(*ν*) is the absorbance, *S*(*T*) (cm/mol) is the line strength, which is dependent on temperature only, *g*(*ν*) (cm) is the normalized line-shape function, *N* (mol∙cm^−3^) is the number density of the test molecules. The line-shape function depends on temperature, pressure, gas composition, and the mechanisms of line broadening.

The line strength *S* depends on temperature and can be expressed as:(2)$$S\left(T\right)\approx S\left(T_{0}\right)\frac{Q\left(T_{0}\right)}{Q\left(T\right)}exp\left[-\frac{hcE^{"}}{k}\left(\frac{1}{T}-\frac{1}{T_{0}}\right)\right]$$
where *T*_0_ is a reference temperature, *Q*(*T)* is the partition function, which depends only on temperature, *E*'' is the lower state energy of the quantum transition, *k* is Boltzmann’s constant, *S*(*T*_0_) is the line strength at a reference temperature *T*_0_, *h* is Planck’s constant and *c* is the speed of light. While the integral over the frequency *ν* of the *g*(*ν*) is equal unity, the integrated absorbance *A_j_* of transition *j* can be expressed for uniform distribution in the form: (3)$$A_{j}\left(T\right)=\int_{-\infty }^{\infty }\alpha_{j}\left(\nu \right)d\nu =\overset{L}{\underset{0}{\int }}S_{j}\left(T\right)Ndl=S_{j}\left(T\right)NL$$

Gas temperature can be determined by the measurements of the absorption on two transitions with different energies of lower levels. For a medium with uniform distribution of temperature and absorbing molecules concentration, the integrated absorbance ratio *R* depends only on temperature and does not depend on concentration *N* or optical length *L*: (4)$$R=\frac{A_{2}}{A_{1}}=\frac{S_{2}\left(T\right)}{S_{1}\left(T\right)}.$$

Hence, the gas temperature can be deduced from the ratio of the integrated absorbance of two lines. 

Errors in temperature measurement usually depend on the accuracy of measuring the parameters of the probe DL radiation and on the accuracy of the fitting process. In real industrial and experimental installations, there are many reasons that limit this accuracy. These issues are discussed further when describing the measurement and data processing methods.

In the present publication, the log-WMS-1*f* technique [31] is used for two DLs to minimize errors in measuring the temperature of a hot zone. For example, consider the algorithm of the integral absorbance evaluation for one line.

If the DL current is modulated with a frequency *f*, then the laser intensity $I_{0}\left(t\right)$ and instantaneous laser frequency *ν*(*t*) can be described as: (5)$$I_{0}\left(t\right)=I_{slow}+a\times \cos\left(2\pi ft+\psi_{1}\right)+b\times \cos\left(4\pi ft+\psi_{2}\right)$$
(6)$$\nu =\nu_{slow}+a_{m}\times \cos\left(2\pi ft\right)$$
where the subscript “slow” defines the values (*I* or ν), which vary slowly during the scanning of the DL frequency; *a* is the laser intensity modulation amplitude (linear part); *b* describes the non-linearity of laser intensity modulation; *ψ*_1_ and *ψ*_2_ are the phase shifts between the modulation of the optical frequency and modulation of laser intensity for the first and second harmonics, respectively; and *a_m_* is the optical frequency modulation amplitude.

In the sample channel, the main part of the DL radiation is delivered to the object under test. Part of the radiation is directed to a reference channel that does not contain an absorbing medium. The signals from the photodetectors of the sample and reference channels are transmitted to the LCs and then to a differential amplifier. Assuming that the photodiodes operate in the linear region of the dynamic range, the photocurrent *i_s_* in the sample channel can be written as:(7)$$i_{s}=G_{s}\left[I_{0}\left(t\right)\times \tau_{s}\left(t\right)\times \tau_{\nu s}\left(t\right)\times \exp\left(-\alpha \right)+E\left(t\right)\right]+i_{noise},$$
where *G_s_* (A/W) is the responsivity of the photodiode, *τ*_s_(*t*) and *τ*_*ν*s_ are the non-selective and selective (excluding absorption) components of the transmission coefficient of the sample channel, respectively; *E(t)* is the intensity of broadband radiation falling on the photodiode, *i_noise_* is non-radiation-related noise. As a rule, the selective component of **τ*_ν__s_* is determined by interference effects on the optical components.

It can be seen from expression (7) that the accuracy of the absorption coefficient measurement is affected by fluctuations in the laser intensity and broadband radiation, as well as uncertainty and instability of the transmittance. At high laser radiation intensities and with the use of suitable spectral and spatial filtering, the influence of noise current and broadband radiation can be neglected. This leads to a simplification of expression (7) for the photocurrent of the sample channel:(8)$$i_{s}=G_{s}\left[I_{0}\left(t\right)\times \tau_{s}\left(t\right)\times \tau_{\nu s}\left(t\right)\times \exp\left(-\alpha \right)\right]$$

A similar expression can be written for a reference channel in which there is no selective absorption:(9)$$i_{r}=G_{r}\left[I_{0}\left(t\right)\times \tau_{r}\left(t\right)\times \tau_{\nu r}\left(t\right)\right]$$

After passing the signals through the logarithmic converter (LC) voltages *U_s_*, *U_r_* at the outputs of the sample and reference channels, respectively, as well as the differential voltage *U_dif_* are described by the following expressions:(10)$$U_{s}=L_{c}\left\{lnG_{s}+\ln\tau_{s}+\ln\tau_{\nu s}+\ln\left[I_{0}\left(t\right)\right]-\alpha \right\},$$
(11)$$U_{r}=L_{c}\left\{lnG_{r}+\ln\tau_{r}+\ln\tau_{\nu r}+\ln\left[I_{0}\left(t\right)\right]\right\},$$
(12)$$U_{dif}=U_{r}-U_{s}=L_{c}\left(\ln\frac{G_{r}}{G_{s}}+\ln\frac{\tau_{r}}{\tau_{s}}+\ln\frac{\tau_{\nu r}}{\tau_{\nu s}}\right)+L_{c}\alpha ,$$
where *L_c_* is the coefficient of logarithmic conversion.

The first term in parentheses of Equation (12) is constant; the second term fluctuates with characteristic frequencies, usually located in the range from zero to several tens of kilohertz. In an optimally designed installation, the third term, responsible for interference fringes, can be significantly reduced. 

The absorbance *α* can be expanded in the Fourier series in the form:(13)$$\alpha \left(\nu \right)=\alpha \left(\nu_{slow}+a_{m}\times \cos\left(2\pi ft\right)\right)=\overset{\infty }{\underset{0}{\sum }}H_{n}\times \cos\left(2\pi nft\right)$$

The absorbance is recorded using a lock-in amplifier at the first harmonic of the modulation frequency. If the laser radiation is modulated at frequencies of several tens of kilohertz and higher, then the output signal of the logarithmic converter *U*_1*f*_ (*ν*) at the modulation frequency becomes proportional to the first harmonic *H*_1_(*ν*) of the Fourier expansion of the absorption coefficient.
(14)$$U_{out,1f}\left(\nu \right)=L_{c}H_{1}\left(\nu \right)$$
where:(15)$$\;\;\;\;\;\;\;\;\;\;\;\;\;\;\;H_{1}=\frac{1}{\pi }\overset{\pi }{\underset{-\pi }{\int }}\alpha (\nu_{slow}+a_{m}\cos\varphi )\cos\varphi d\varphi$$

Having measured the amplitude $a_{m}$ of the frequency of the laser radiation modulation, it is possible to evaluate α and integral absorbance *A* by fitting the simulated spectra *H*_1_(*ν*) and experimental log-WMS-1*f* spectra. From simultaneous measurement of the integrated absorbance of two lines one can evaluate the temperature of an object Equation (4) and from the measurement of the absolute integrated absorption line intensity deduce the concentration of the absorbing species for the length of the probing beam in the medium *L*, provided that the spectroscopic parameters of the selected transitions are known Equation (1). 

## 3. Experimental

### 3.1. Set-Up

The experimental set-up is presented in Figure 1.

Atmospheric air with water traces was flushed in the central section (8 cm long) of a heated quartz tube. A quartz tube, 30 cm long and 50 mm i.d., was located inside the electrically heated cylindrical furnace. The central section was formed between quartz wedge-shaped windows of two inserts, each 30 cm long and 40 mm i.d. A small angle between the planes of the windows minimized interference between the planes. The temperature in the central section was controlled by three commercial thermocouples PS2007 Instrument Specialists Inc., Boerne, TX, USA. The accuracy of the thermocouples was 7.5 K at 1000 K. The length of the central section was about four times less than the length of the oven providing practically uniform temperature of the gas within the section, the variation of the temperature at 1000 K was within +/− 0.3%.

Two DLs were used to probe the transitions of the H_2_O molecule at 7185.6 (DL1, NEL709042) and 7466.34 cm^−1^ (DL2, Zacher Lasertechnik 1343-05-BFY). Both DLs were stabilized and tuned by temperature (TED350) and current (LDC202) controllers (Thorlabs). A signal of a special form was additionally applied to the inputs of both controllers. This signal provided low frequency (LF) scanning of the DLs wavelengths and high frequency (HF) modulation of both DLs. The amplitudes of both LF and HF signals were optimized for both lasers independently, but LF and HF frequencies for both lasers were equal 122 Hz and 50 kHz. 

The radiation of both DLs were combined in the single-mode *sm* splitter 1 and the mixed radiation was further split into three channels by the 3-output single-mode *sm* splitter 2. The first (interference) channel (3% of the radiation) was used for controlling the DLs modulation amplitude and linearization of the wavelength scale. To control the DLs scanning the radiation was delivered by the fiber *s1* to the fiber Mach–Zander interferometer (etalon) with a free spectral range 0.0171 cm^−1^. A total of 7% of the output radiation was delivered to the photodiode PDr of the second (reference) channel by fiber *s2*. The remaining 90% of the radiation formed the sample channel. The output of the single-mode fiber *s3* was attached to the input/output of the multimode fiber *m2*. Then DLs radiation passed through the *mm* splitter, a 10 m long multimode patch cord *m1* and was finally collimated by the objective on a Thorlabs F240APC-C optical collimator. The probing radiation passed through the gas cell, reflected from the micro-prism retroreflector (MPRR), focused by the same lens onto the face of fiber *m1*, and was delivered to the photodiode of the sample channel via the *mm*-splitter and fiber *m3*. The photodiodes Hamamatsu G8370-02 (PDi, PDr, and PDs) with a sensitive area diameter of 2 mm were used in all three channels. 

The acoustic fluctuation and turbulence on the optical path of the probing DLs radiation complicated the focusing of the laser beam onto the input/output of the fiber *m1*. To reduce these fluctuations, the probing beam was reflected back to the detection system by micro-prism retroreflector MPRR [31]. The multimode optical fiber *m1* with a 50 μm core was also used to increase the collection efficiency of the reflected probing DL beams. At the same time, its aperture was small enough to minimize the broadband thermal radiation of the hot zone of the furnace. The efficiency of the reflected radiation collection to the sample photodiode PDs was about 1%. Such low efficiency was nevertheless enough for processing the absorption signal. At the same time, the negative feedback of the reflected radiation on the performance of the DLs was negligible. The measured feedback light intensity was less than 0.1%. MPRR and focusing optics were placed in a separate box and flushed with the argon. More details on the performance of the MPRR are published elsewhere [31]. 

A rotating plastic disk with a specially roughened surface, which randomly modulated the laser beam transmitted intensity, was used to simulate multiplicative noise. A DC motor rotated the disk at various speeds of up to 3000 rpm.

Signals from the sample and reference channels were delivered directly to the analog logarithmic converters LC1 and LC2, based on the *p-n* junctions of the MAT-04 transistors (Analog Devices, subpanel in Figure 1). The amplified differential signal was demodulated by a homemade lock-in-amplifier LIA, based on an Analog Devices AD633 analog multiplier. The signal from the LIA was filtered by a low-frequency filter with a passband frequency of 3 kHz and digitized via a National Instruments NI USB-6281 data acquisition (DAQ) system.

### 3.2. Measurement Procedure

At first, measurements were carried out in a direct absorption mode. To do this, the transistor MAT04 was replaced with a 6.2 kΩ resistor and the signal was measured at the output of the operational amplifier AD8034 (see insert in Figure 1). The modulation of the lasers was switched off. By adjusting the optical scheme, MPRR, and focusing the spherical mirror, a maximum value of the photocurrent in the measuring channel and the minimum interference effect were achieved. At a temperature of about 1000 K, the absorption signal on the 7466 cm^−1^ line became large enough, and it could be additionally increased by introducing moist air into the central part of the cell. Then the absorption signals and the transmission spectrum of the Fabry–Perot etalon were measured simultaneously. These data were used for the linearization of the wavelength scale for both 7185 and 7466 cm^−1^ spectral ranges.

Next, the background signal, defined by the broadband thermal radiation of the cell, backscattered radiation from the *mm*-fiber, and the noise in the electrical circuit, was measured with the DLs switched off. In the described system the additive signal was small enough to provide correct logarithmic processing.

At the next step, the scanning of both DLs was switched off and the signal transmitted through the cell was measured at constant DL radiation intensities. In these experiments, the output signal was measured in two modes: with the rotating disk off and on. These data were used for the evaluation of the noise spectrum in the sample channel. All further measurements were performed with the scanning and modulation of the DLs switched on. The modulation amplitude of DL2 was adjusted to maximize the signal (peak-to-peak) of the first harmonic. The amplitude of modulation of DL1 was adjusted to approximately equalize the signals of the first harmonic absorption signal registered by both DLs (7185 and 7466 cm^−1^). This adjustment provided the correct linear performance of the LIA. 

Prior to the measurements at every temperature of the cell, the optical path was flushed with argon for ~5 min. The residual water vapor in the system even after 10 minutes of argon flush, provided an absorption signal at the 7185 cm^−1^ line above the noise level. This minimal level of 7185 signal was registered and further processed as the background signal. After registration of the background signals on both lines, ambient air was introduced into the cell and the first harmonics of the absorption signals at both lines were detected in the stationary mode. Importantly, to localize the water vapor absorption only in the cell, all parts of the flushed optical path excluding the measurement cell were maintained at the same constant level. Just after these measurements with the plastic disk off, the disk was inserted into the optical path and the measurements with the added multiplicative noise were performed. 

## 4. Results and Discussion

### 4.1. Influence of Broadband Radiation

When deriving Equation (12), we neglected the influence of the broadband illumination *E*(*t*) of the photodetector, which can be significant when working with hot objects. The presence of additional thermal radiation leads to the appearance of an additive term in the photocurrent of the sample channel:(16)$$i_{s}=i_{sig}+i_{add}=i_{s0}\exp\left(-\alpha \right)+i_{add}.$$

Here, $i_{sig}$ is the photocurrent of the sample channel when the broadband illumination is negligible; *i_s_*_0_ = *G_s_*τ*_s_*(*t*)*τ*_*ν*s_(*t*)*I*_0_(*t*); *i_add_* is the photocurrent due to the broadband illumination; *i_add_ = G_s_E(t).* In this case, the output voltage of the logarithmic converter in the sample channel is described by the equation:(17)$$U_{s}=L_{c}\ln(i_{sig}+i_{add})=L_{c}\ln i_{sig}+L_{c}\ln(1+\frac{i_{add}}{i_{sig}})$$

As a result, an additional voltage *U_add_* appears at the output of the differential amplifier, which for small absorptions is described by the equation: (18)$$U_{add}=-L_{c}\ln(1+\frac{i_{add}}{i_{sig}})\approx -L_{c}\frac{i_{add}}{i_{sig}}\approx -L_{c}\frac{i_{add}}{i_{s0}\exp(-\alpha ))}\approx -L_{c}\frac{i_{add}}{i_{s0}}-L_{c}\alpha \frac{i_{add}}{i_{s0}}$$

It can be seen from Equation (17) that the illumination of the photodetector by the broadband radiation leads to a shift of the baseline at the output of the lock-in amplifier and to a decrease in the measured absorption signal by an amount proportional to *i_add_/i_s_*_0_. Additional illumination can also lead to errors in determining the temperature, since the values of the signal currents *i_s_*_0_ in the vicinity of the two spectral lines can differ significantly. Therefore, the requirements for shielding the sample photodiode from broadband illumination become critical.

In our work, the radiation was collected in the sample channel through a multimode optical fiber with a core diameter of 50 μm. Thus, the input aperture in relation to broadband illumination was radically reduced and, accordingly, the contribution of *i_add_* was reduced. For reliable focusing of laser radiation on the face of the fiber, the principle of autofocus using a micro-prism retroreflector was applied. 

As was pointed above the errors of the logarithmic processing depend on the level of the additive component and are approximately defined by Equation (17). To check this claim, scanning (122 Hz) and modulation (50 kHz) of DL1 were switched on and the first harmonic of the absorption signal on the 7185 line was measured with logarithmic processing. At the same time, DL2 worked in a steady state mode and its radiation provided the additive input in the photocurrent of the sample photodiode PDs. The level of this additive component was maintained in the linear response mode and could be varied. The results of these measurements, presented in Table 1, confirm the validity of Equation (17). Based on the data in Table 1, one can conclude that the additive component of the current through the LC should be minimized. 

In our experiments the additive contribution in the signal at 1000 K was below 3 × 10^−3^. This contribution was defined by the photocurrent from the broadband thermal radiation, the reflection of the DL radiation from the non-ideal connections in the *mm*-splitter, and the leakage current of the input electronic circuit cascades. 

### 4.2. Errors of Non-Ideal LC

The use of LCs based on *p-n* junctions may introduce additional sources of uncertainties. During the modulation experiments, the errors of logarithmic processing increase with increasing modulation frequency because of leakage through the shunt capacity of the *p-n* junction. This capacity is defined by the eigen capacity of the junction, the input capacity of the operational amplifier, and the effective capacity of the photodiode. The capacity of a photodiode with a diameter of a sensitive area of 2 mm can be up to 1000 pF. The dynamic resistance of a *p-n* junction is defined by its current, and for currents less than 10 μA, this can be above several kΩ. As a result, the relation between high-frequency and low-frequency components at the output of the LC decreases compared to the input. In addition, a phase shift occurs in the high-frequency component. This effect increases with increasing modulation frequency. Significant deviations of the LC parameters from the ideal may cause additional errors in the determination of spectroscopic line parameters and, consequently, introduce errors in the determination of a gas temperature. To minimize these errors, one should use photodiodes with small eigen capacities, for example, photodiodes integrated with fibers and the radiation collection efficiency should be increased using high-quality MPRR. In our set-up, the moderate quality of all components dictated the selection of a limited modulation frequency −50 kHz. Another reason for the limitation of the modulation frequency is low efficiency of the wavelength tuning of the DLs, namely, a decrease of the modulation amplitude with increasing modulation frequency. Because of this limited efficiency, the level of RAM increases and approaches the maximum permitted level of a DL injection current.

### 4.3. Characteristics of Excess Multiplicative Noise

When using a rotating plastic disk (Figure 1), the mean level of photocurrent decreases by ~20% and simultaneously fluctuations sharply increase. Spectra of the photocurrent noise with (disk on, black trace) and without (disk off, red trace) the multiplicative components, measured in the direct mode are presented in Figure 2. The low frequency part of the noise spectrum with multiplicative components (black) increases by 45 dB compared to the spectrum without multiplicative components (red) and follows a 1/*f* dependence. Excess flicker noise exists below 60 kHz.

The efficiency of the new data processing algorithm is shown in Figure 3. The figure shows 3D images of raw data (signals from the photodetector PDs) registered in the DAS mode (a) and using the log-WMS-1*f* technique (b) for a temperature of 1050 K.

The 3D spectra registered in DAS mode exhibit well-defined absorption lines in both spectral ranges when there were no extra noises (disk off), but weak lines in the 7466 cm^−1^ spectral range are indistinguishable in noise with the disk on. On the contrary, absorption lines in both spectral ranges are well detected even with extra noise using the log-WMS-1*f* technique.

The same results were exhibited in the absorption spectra detected in a single scan. Raw data for single-scan measurements are shown in Figure 4. Signals detected in DAS mode are shown in the left panels; signals detected in the log-WMS-1*f* mode are shown in the right panels. The upper traces in both panels were detected without extra noise in the sample channel (disk off), while the lower traces were detected with extra noise (disk on). Large differences in the efficiency of the absorption spectra registration in the two modes were evident. The log-WMS-1*f* technique provided the efficient elimination of extra multiplicative noise and enables sensitive registration of weak absorption lines. 

### 4.4. Temperature Measurements

Several series of measurements were performed at temperatures ranging from 800 to 1100 K. In each series, the temperature was measured by a thermocouple and by the spectroscopic technique. The log-WMS-1*f* spectra were recorded without a rotating plastic disk (disk off) and with the insertion of additional multiplicative noise (disk on). At each temperature, 18 scans were recorded in one measurement. Each scan was processed separately, as well as averaged (over 18 scans) for one measurement. The amplitudes of modulation of the optical frequency of the lasers varied from series to series. Below are the results for a series in which the modulation amplitudes were set to *a_m_*_1_ = 0.0057 cm^−1^ for the 7185 cm^−1^ laser and to *a_m_*_2_ = 0.017 cm^−1^ for the 7446 cm^−1^ laser.

To determine the temperature, the measured spectra were fitted to a theoretical spectrum *H*_1_(*v*). The absorption lines were constructed assuming a Voigt profile. Before the fitting procedure, the experimental spectra (raw data) were transformed to the frequency domain (cm^−1^) using the measured spectrum of the Fabry–Perot etalon.

Two processing algorithms were used. The flow chart for the first algorithm is shown in Figure 5. In this algorithm, two spectral ranges (7185 and 7466 cm^−1^) were fitted as a single spectrum. Initially, the spectrum of the first harmonic was simulated for a certain starting (guess) temperature for each section from the HITRAN database [32] according to Equation (15). The Doppler widths were fixed at a guess temperature. Fitting the simulated spectrum to the experimental one was carried out using the nonlinear least squares method. Least squares (model fitting) algorithms [33] were employed. In each range, the fitting parameters were the positions of the centers of the absorption lines ν_01_, ν_02_, and their Lorentzian widths Δν*_L_*_1_ and Δν*_L_*_2_. The common fitting parameters in the two ranges were temperature *T* and coefficient *K*. As suggested in [34], the contribution of the first three orthogonal polynomials was additionally subtracted from the experimental and simulated spectra. From a mathematical point of view, this was equivalent to fitting the baseline of the experimental spectra using a parabola [35].

The flow chart for the second fitting algorithm is shown in Figure 6. In this algorithm, the simulated spectrum *H*_1_(*v*) was fitted separately for each spectral range. At the initial (guess) temperature *T_g_*, the line absorbance *A*_1_ and A_2_ were determined independently for the position of the line centers ν_01_ и ν_02_, the Lorentzian widths of the lines Δν*_L_*_1_, Δν*_L_*_2_, and the coefficient K. Temperature *T* was inferred using Equation (4). If *T* was noticeably different from *T_g_*, then an iterative procedure was started. Iterations were stopped when the difference in temperature evaluation between two successive steps was less than 10 K. This temperature was assumed as the measured gas temperature in the cell.

As an example, Figure 7 shows the spectra measured in one scan for *T* = 1050 K (thermocouple) and the residuals from processing using the first algorithm. The developed log-WMS-1*f* technique provides a very reasonable estimation of the gas temperature, even in one scan. Note that the laser was scanned with a frequency of 122 Hz, which gives an estimation of the temporal resolution of about 8 ms. The temperature obtained as a result of fitting was *T* = 1022 K for the disk off. For the experiments with additional noise (disk on) the temperature inferred from the “noisy” spectra was *T* = 1040 K. Fitting using the second algorithm gave 1043 and 1057 K, respectively. 

The results of fitting of all scans from this series of experiments for temperature *T* = 1050 K (thermocouple) are shown in Figure 8. Each point in Figure 8 is the *T* value obtained in a particular scan when processed by two algorithms and with the disk off/on (see the legend to the figure). In some scans, the deviations from the temperature of the thermocouple were larger with noise, and in some without noise. The values for three temperatures averaged over 18 scans and according to statistical errors for different modes (disk off/on) and different processing algorithms are presented in Table 2. These results show that statistical errors in the case of noise were greater than those without noise. Nevertheless, due to the good noise suppression by our proposed method, the temperature estimate was quite good even in the presence of excessive noise.

Experiments were performed with the plastic disk off (without extra noise) and on (with extra noise). The temperature was evaluated using two algorithms of data processing. For all series of experiments conducted the difference between the temperature measured by a standard thermocouple and the average value determined by the developed methods was, with the disk turned on, Δ*T* ≤ 40 K for temperature ~1000 and Δ*T* ≤ 30 K for temperature ~800 K. 

The results presented in Table 2 show that both algorithms of data processing provided quite close temperature estimates. Generally, the second one was better for the case of the large difference in the intensities of absorption lines, while in the process of separate fitting the strong line did not “impose” its line shape on the weak one. In our review [35], we discussed the situation with lines of comparable intensity when both algorithms were equivalent. In the current paper, the lines were very different, and it was better to use the second algorithm. The first algorithm can be used in the case of registration of both lines within the tuning range of one DL [9,13]. In this situation, the simultaneous fitting of lines of comparable intensity will allow a better approximation of the baseline.

## 5. Conclusions

The potential of a new version of a TDLAS sensor based on the log-WMS-1*f* method was demonstrated by measuring temperatures in a laboratory cell with stable and variable conditions. This version was based on the modulation of two diode lasers with a frequency of about 50 kHz, logarithmic conversion of the signals, and detection of the first harmonics of the absorption signals. The temperature was evaluated from the ratio of integrated absorbance of two lines of the water molecule with different lower energy levels. Two algorithms of data processing were used—independent fitting of the two absorption line profiles, and simultaneous fitting of both spectral ranges with two lines. The influence of extra non-selective noise on the result of temperature evaluation was investigated. Extra multiplicative noise was artificially simulated by introducing in the probing DLs path a rotating disk with rough surfaces. A combination of logarithmic conversion, signal detection by LIA, and a differential scheme of data processing greatly minimized multiplicative noise and improved the accuracy of gas temperature estimates. Under the condition with extra noise, temperatures evaluated by the proposed technique differed from the gas temperature measured by a commercial thermocouple by less than 40 K for a temperature of ~1000 K and less than 30 K for a temperature ~800 K. The developed technique enabled a reasonable evaluation of a gas temperature (Δ*T* ≤ 40 K) even in a single scan of DL wavelength, which provided a time resolution of temperature estimation of about 8 ms. The developed log-WMS-1*f* technique is an alternative to the calibration-free 2*f/*1*f* -WMS. The advantages of the log-WMS-1*f* compared to 2*f/*1*f*-WMS are: simpler data processing; a higher level of 1*f* signal over 2*f* signal, which means higher sensitivity; larger linear dynamic range for measuring the absorption signal, and simpler instrumentation. The developed log-WMS-1*f* technique has good potential for applications in harsh environments.

The drawback of the proposed scheme is the analog processing of the signals. It limits the modulation frequency of DLs because of increasing parasitic capacities. 

In recent years, the development of new instrumentation has led to increasing use of digital signal registration in TDLAS. To isolate the harmonics of the signal, both commercial digital LIAs and LIAs based on LabVIEW are used. In the WMS-2*f*/1*f* technique, the 2*f* and 1*f* signals are measured and digitized, then 2*f*/1*f* division is fulfilled and, finally, the quotient of the division is fitted. With a high level of multiplicative fluctuations additional difficulties arise if the fluctuating denominator is small. In our method, the output signal filtered from noise is immediately digitized and fitted. This simplifies the procedure.

In a method using digital registration with Fourier signal processing, good amplitude and time resolution of DAQ are necessary in order to measure weak absorption signals at the first and second harmonics against the background of a large base signal and fluctuations. This is not a problem with single-channel measurement [36], but greatly complicates the equipment for multichannel (several dozen) simultaneous measurements, as is necessary, for example, for tomography of inhomogeneous gas flows [37]. In the proposed scheme, the requirements for amplitude resolution and the sampling rate of DAQ are much weaker, and multichannel measurements will not cause serious problems.

## Figures and Tables

**Figure 1 sensors-23-00622-f001:**
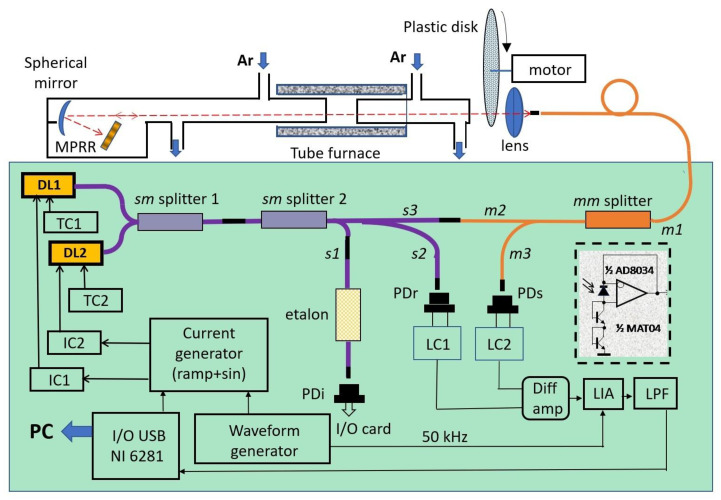
Experimental set-up. MPRR—micro-prism retroreflector; DL1, DL2—diode lasers; TC, IC—temperature and current controllers, respectively; *sm*—single mode, *mm*—multimode; *s1*, *s2*, *s3*—single-mode fibers; *m1*, *m2*, *m3*—multimode fibers; PDr, PDs, PDi—photodiodes of the reference, sample, and interferometric channels, respectively; LC1, LC2—logarithmic converters of the reference and sample channels, respectively; Diff amp—differential amplifier; LIA—lock-in amplifier; LPF—low-pass filter; PC—personal computer. The electrical scheme of the LC is shown in the insert.

**Figure 2 sensors-23-00622-f002:**
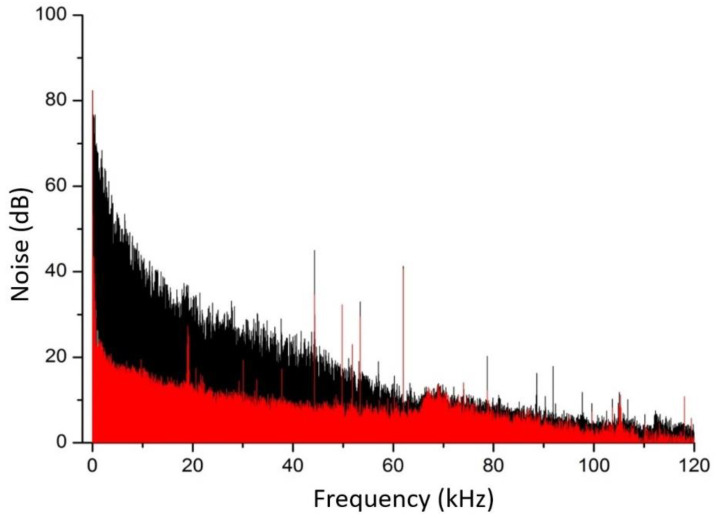
Spectra of the photocurrent noises; plastic disk off (red trace), plastic disk on (black trace).

**Figure 3 sensors-23-00622-f003:**
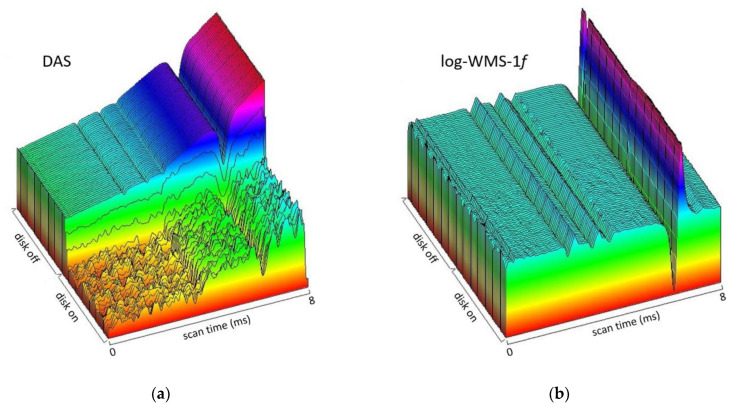
Signals from the photodetector registered in DAS mode (**a**) and log-WMS-1*f* mode (**b**)**.**

**Figure 4 sensors-23-00622-f004:**
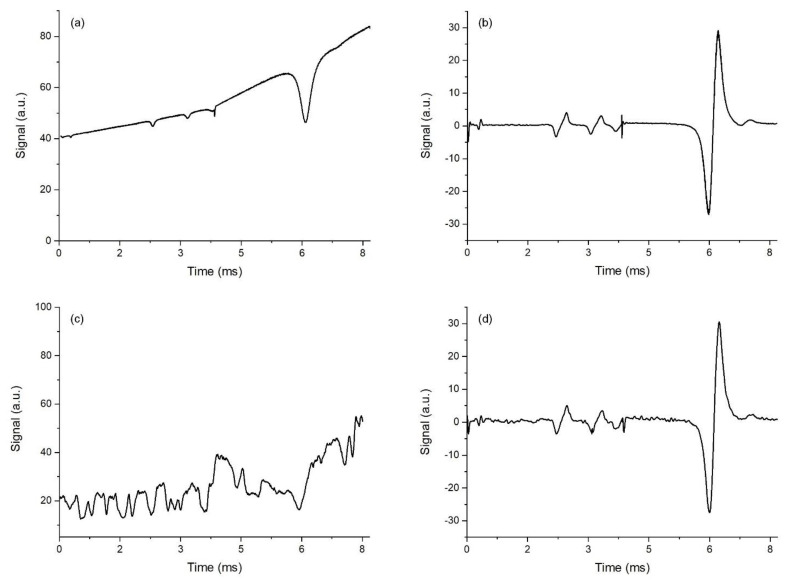
Raw data of a single scan detected in DAS mode (left panels) and log-WMS-1*f* mode (right panels); disk off—(**a**,**b**); disk on—(**c**,**d**).

**Figure 5 sensors-23-00622-f005:**
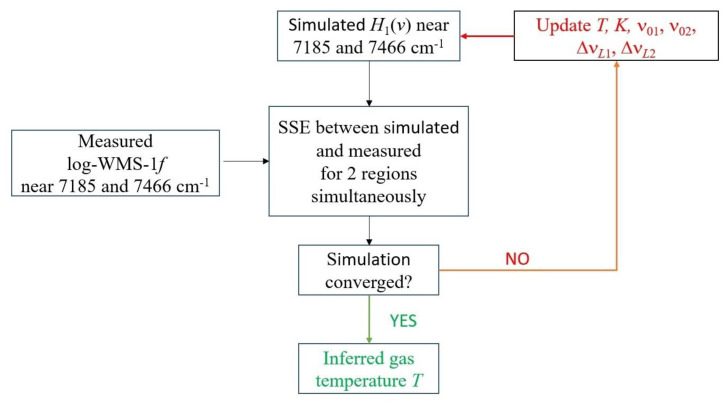
Flow chart for the first algorithm for inferring the temperature from the measured spectra. SSE is the sum of squared errors.

**Figure 6 sensors-23-00622-f006:**
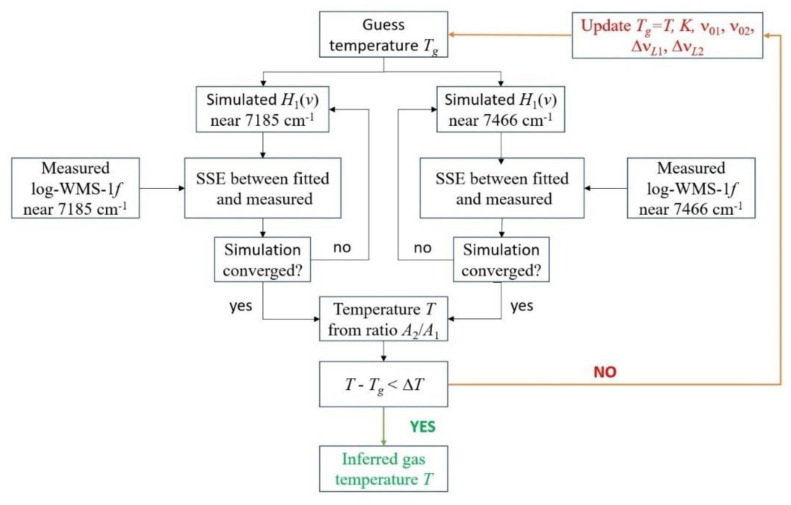
Flow chart for the second algorithm for inferring the temperature from the measured spectra.

**Figure 7 sensors-23-00622-f007:**
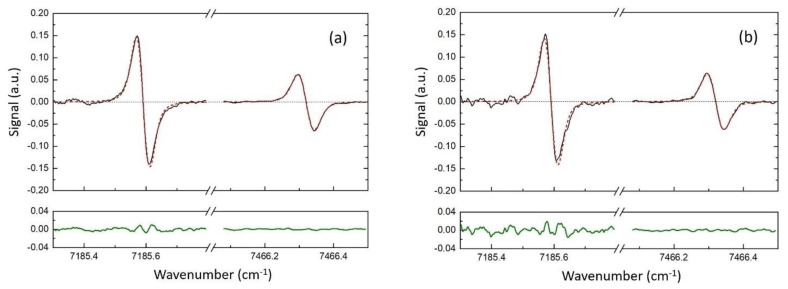
Fitting of the spectra measured by the log-WMS-1*f* technique in one scan with rotating disk off (**a**) and disk on (**b**). The measured spectra—black lines; the best-fit simulated spectra—red lines; residuals—green lines.

**Figure 8 sensors-23-00622-f008:**
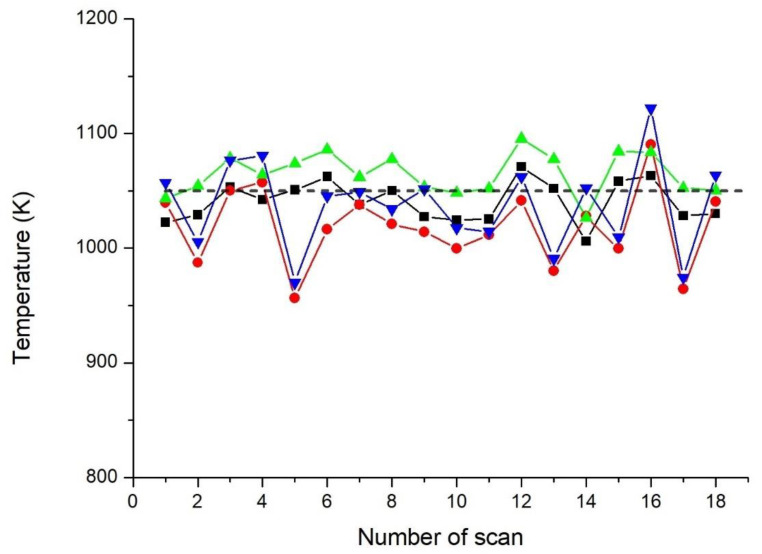
The results of temperature evaluation by the log-WMS-1*f* technique in each of the 18 scans. The dotted black trace is the reading of the thermocouple. Plots with symbols are inferred temperature from log-WMS-1*f* single scans: solid black plot—disk off, fitting by algorithm 1; green plot—disk off, algorithm 2; red plot—disk on, algorithm 1; blue plot—disk on, algorithm 2.

**Table 1 sensors-23-00622-t001:** Dependence of the peak-to-peak signal of the first harmonic *U_ptp_* on the relative contribution of the additive current *i_add_*.

$\frac{i_{add}}{i_{sig}}$	$\frac{{\left(U_{ptp}\right)}_{max\;\;}-{\left(U_{ptp}\right)}_{measured}\;}{{\left(U_{ptp}\right)}_{max\;\;}}$
0	0
8 × 10^−3^	0.01
4.8 × 10^−2^	0.05
8.8 × 10^−2^	0.085

**Table 2 sensors-23-00622-t002:** Results of temperature evaluation by the log-WMS-1*f* technique.

Thermocouple Temperature (K)	Log-WMS-1*f* Temperature (K)			
	Disk off		Disk on	
	Algorithm 1	Algorithm 2	Algorithm 1	Algorithm 2
1050	1040 ± 17	1064 ± 18	1019 ± 34	1035 ± 39
885	913 ± 10	931 ± 11	898 ± 43	908 ± 44
815	834 ± 10	846 ± 12	810 ± 39	811 ± 42

## Data Availability

Not applicable.

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
