# Peer review of "Temperature Measurements by Wavelength Modulation Diode Laser Absorption Spectroscopy with Logarithmic Conversion and 1f Signal Detection"

_sensors, 2023, doi:10.3390/s23020622_

Round 1

Reviewer 1 Report

A method of high temperature measurements by wavelength modulation diode laser absorption spectroscopy with logarithmic conversion and 1f signal detection is studied. The research with clear logic and detailed content is well presented. This is a very meaningful work, which can be used for reference by relevant readers.

It is recommended to accept after modification.

1. In the introduction, the authors explain the motivation of this article very clearly, but since there are many studies related to TDLAS temperature measurement, it is recommended to add relevant references

2. Figure 4 suggests adding the 1f signal without logarithmic processing of the original signal for comparison with the 1f signal with logarithmic processing.

3. Line 453 of page 13, please explain the temperature measured without noise has a larger error than the temperature measured with noise. One measurement is accidental. How about repeated measurements?

4. Figure 8 shows a small amount of temperature data, so it is suggested to add long-term measurement data. And in Tab.2, results of temperature evaluation by the log-WMS-1f technique are presented, how many measurements are the mean value and standard deviation obtained based on?

5. What is the roughness of the plastic disk? Will there be speckle effect?

6. The variables in the text should be italicized and their meanings should be defined, such as isig and iadd in Eq.(16).

Reviewer 2 Report

The authors developed a new WMS so call log-1f-wms for temperature measurements. The novelty is using the logarithmic conversion of the 1f-wms signal to reduce the multiplicative noise effects. The paper is well written. There are some minor comments: 

1, Figures 4, 7 and 8, they should be in the same format.

2, In Figures 5 and 6, there are two different algorithms, please provide detailed information on the pros and cons of each algorithm, and which algorithm should be used in what kind of conditions. 

3, The method developed in this paper has been used for temperature measurement, the author should also provide a view when it comes to concentration measurements, which will be very helpful to the readers. 

 4, In conclusion, the authors claim 'log-WMS-1f compared to 2f/1f -WMS are: simpler data processing; a higher level of 1f signal over 2f signal, which means higher sensitivity; larger linear dynamic range for measurement the absorption signal and, simpler instrumentation.'  There are several calbration-free 2f/1f -WMS which are based on digital LIA, you can't claim here simple data processing and instrumentation. 

5, The last sentence in Introdection: 'The advantages and limitations of the proposed technique are discussed.' It seems the limitations of the technique is missing in current version. 

Reviewer 3 Report

The paper Temperature measurements by wavelength modulation diode laser absorption spectroscopy with logarithmic conversion and 1f signal detection by Liger et al. report a temperature measurement system based on tunable diode laser absorption spectroscopy. They used the ratio of the two absorption spectra of water vapor to evaluate the temperature. My comments are as follows:

a. In line 64, as far as I know, the scanning frequency of wavelength modulation technology cannot be too high in some applications, such as photoacoustic spectroscopy, which may be only a few Hz or even lower. In addition, please clarify the modulation frequency range of wavelength modulation technology is 10 Hz-100 kHz, or 10 kHz-100 kHz ?

b. Formula 11 seems wrong, and τs should be τr. Please check minor errors of the paper.

c. In my opinion, readers can 't understand the description of lines 211 to 231 very well. Please add relevant content or size details in Figure 1.

d. The abscissa in Figure 4 is better replaced by wavenumber.

e. The effects of the two processing algorithms on the temperature detection accuracy seem to be the same. Please explain the advantages and disadvantages of the two algorithms in the manuscript. If the advantage is not obvious, I think the author can only retain a simple algorithm.

f. I think the descriptions in Figure 8 and Table 2 are duplicates, so keep only one.

g. Compared with thermocouples, the detection accuracy of this method has not been improved, and the cost has not been reduced. Please explain the advantages of this method, such as non-contact measurement.

h. The author 's experiments are carried out at high temperature, which is easy to make readers wonder. Please supplement the experiment at low temperature, or explain the reason why the method cannot be applied to low temperature in the manuscript.

Round 2

Reviewer 3 Report

Authors gave answers to comments and suggestions; I consider the manuscript can be accepted for publication this time.